



# The Radiative Forcing Model Intercomparison Project (RFMIP2.0) for CMIP7

Ryan J. Kramer[1], Chris Smith[2,3], Timothy Andrews[4,5]

[1]Geophysical Fluid Dynamics Laboratory, NOAA, Princeton, NJ, USA
[2]Department of Water and Climate, Vrije Universiteit Brussel, Brussels, Belgium
[3]International Institute for Applied Systems Analysis (IIASA), Laxenburg, Austria
[4]Met Office Hadley Centre, Exeter, UK
[5]School for Earth and Environment, University of Leeds, UK

*Correspondence to*: Ryan J. Kramer (ryan.kramer@noaa.gov)

**Abstract.** An external perturbation to the climate system from anthropogenic or natural activity first impacts the climate by inducing a perturbation to Earth's energy budget, known as a radiative forcing. The characteristics of the radiative forcing, such as its global-mean magnitude and spatial pattern, determine the subsequent climate response. Therefore, forming accurate projections of climate change first requires diagnosing radiative forcing and evaluating its persistent uncertainty in Global Climate Models. As part of the Coupled Model Intercomparison Project phase 7 (CMIP7), the second iteration of the Radiative Forcing Model Intercomparison Project (RFMIP2.0) will enable the systematic characterization of effective radiative forcing and its components across state-of-the-art climate models, through a set of fixed-Sea Surface Temperature timeslice and transient experiments. The protocol for RFMIP2.0, introduced here, will in part serve as a continuity and an expansion of core RFMIP experiments first introduced in CMIP6, some of which have now been incorporated into the overarching CMIP7 DECK and FastTrack protocols given their broad utility. This will allow for a consistent estimate for radiative forcing across multiple model generations, which is valuable for model evaluation and future development. RFMIP2.0 also includes new experiments that will address open questions about the definition of radiative forcing, such as its sensitivity to evolving surface conditions, and will further enhance an ever-growing swath of science applications that rely on an understanding of Earth's energy budget.

## 1 Introduction

A change in the composition of the climate system, stemming from natural or anthropogenic activities, can induce a perturbation to the Earth's radiative energy budget known as a *radiative forcing*. Fundamentally, all anthropogenic climate change is a response to a radiative forcing, as the planet attempts to restore energy balance. In its simplest form, the radiative imbalance at the top-of-the-atmosphere $N$ is initially caused by a radiative forcing $F$ and damped by the climate's response.



The response can be collectively represented by a change in global-mean surface temperature $\Delta T_s$, which modifies the imbalance through the climate radiative feedback parameter λ (negative for stability). This process is expressed as:

$$N = F + \lambda \Delta T_s \quad (1),$$


whereby the climate reaches a new equilibrium state when radiative balance is fully restored ($N$=0). Therefore, in order to project how the climate will change in the future, it is crucial to understand and quantify the radiative forcing. This tenet was the key motivation behind the initial establishment of the Radiative Forcing Model Intercomparison Project (RFMIP, Pincus et al. 2016), as designed for the Coupled Model Intercomparison Project (CMIP), and remains so for the next iteration of

RFMIP introduced here.

Although the basic concept described above has held, the precise definition of radiative forcing has evolved since being first specifically introduced by Ramanathan (1975). Today, the metric of *effective radiative forcing* (ERF) is most widely used to represent *F*. The ERF is the sum of the initial, purely radiative *instantaneous radiative forcing* (IRF) and the *radiative adjustments*, which are changes in the climate state, such as in clouds, temperature, water vapor or surface albedo, that are

caused by the IRF but occur independently of global-mean surface temperature change (Myhre et al. 2013; Forster et al. 2021). With the inclusion of radiative adjustments, the ERF is a more useful predictor of surface temperature change than the IRF alone and reduces the need to account for efficacy when comparing the climate impact of different forcings (Sherwood et al. 2015; Richardson et al. 2019; Myhre et al. 2024). While the IRF stems only from a change in radiation and can be computed using standalone radiative transfer algorithms, radiative adjustments additionally involve a change in the climate state.

Therefore, diagnosing radiative adjustments, and thus the ERF, requires a global climate model (GCM) or earth system model (ESM). RFMIP fulfilled this requirement in a systematic manner for CMIP phase 6 (CMIP6; Eyring et al. 2016), contributing to various assessments and an improved understanding of the ERF and its components.

Among other uses, RFMIP experiments have served one of two main purposes: a) the precise quantification of the ERF, its forcing components, and its uncertainties for a specific perturbation, or b) the diagnosis of the temporal evolution of

the ERF and its components as the magnitude of those perturbations change with time (e.g. Smith et al. 2020; Smith et al. 2021a; Forster et al. 2021). Through these applications, RFMIP has helped establish the current state-of-the-art understanding of radiative forcing and its intermodel spread, serving as a key source of information for multiple Working Groups and Chapters of the Intergovernmental Panel on Climate Change 6[th] Assessment Report (IPCC AR6) and in the construction of the IPCC AR6 Effective Radiative Forcing timeseries (Forster et al. 2021; Smith et al. 2021b). Performing robust diagnosis of the

ERF using RFMIP experiments, or mimicking the approach, has become a fixture in the climate model development, evaluation and documentation process (Zhao et al. 2018; Golaz et al. 2022; Mackallah et al. 2022; Govardhan et al. 2023). Likewise, ERF outputs from GCM simulations have become widely used in the construction and tuning of climate model emulators (e.g. Leach et al. 2021).



By enabling the characterization of the ERF time evolution in GCMs, RFMIP experiments are also used in efforts to diagnose and attribute the time evolution of the subsequent *radiative response*. For instance, pairing RFMIP simulations with fully-coupled simulations of the historical period allows one to diagnose the temporal evolution of radiative feedbacks as a residual (e.g. Dong et al. 2021). This has been critical for studying the "pattern effect" - how the spatial distribution of SST trends impacts climate sensitivity (Armour et al. 2024). A similar approach using RFMIP simulations has been applied to satellite-observed changes in Earth's top-of-atmosphere energy imbalance, establishing that anthropogenic radiative forcing has been a key driver of recent trends (Raghuraman et al. 2021, Raghuraman et al.2023; Hodnebrog et al. 2024).

Following equation 1, accurate estimates of the ERF from RFMIP experiments have also enabled improved estimates of the feedback parameter and effective climate sensitivity (e.g. Zelinka et al. 2020). To isolate and quantify these terms, RFMIP fixed-SST simulations with a quadrupling of $CO_2$ concentrations from pre-industrial conditions (*piClim-4xCO2*) are often paired with analogous fully coupled simulations with an abrupt quadrupling of $CO_2$ (*abrupt-4xCO2*). This approach is recommended over regression techniques that use the fully coupled simulations alone (Forster et al. 2016; Pincus et al. 2016). As detailed below, *piClim-4xCO2* will now be included in the suite of baseline DECK (Diagnostic, Evaluation and Characterization of Klima) experiments for CMIP7 (Dunne et al. 202*5*), in part to complement the *abrupt-4xCO2* DECK experiment.

The routine use of RFMIP experiments in many key research applications motivates the development of a second iteration of the project to align with CMIP7 (hereafter RFMIP2.0). The protocol introduced here preserves the core function of RFMIP, allowing for the systematic evaluation of radiative forcing across CMIP7 climate models and a comparison with the radiative forcing from CMIP6 models that participated in the first iteration of RFMIP. In doing so, RFMIP2.0 attempts to address two primary research questions:

1. What is the present-day radiative forcing, and its key anthropogenic contributors, since pre-industrial times?

2. What is the temporal evolution of the radiative forcing, and its components, over the historical period and into the future?

RFMIP2.0 will also be responsive to emerging science that has highlighted the challenges of cleanly defining and diagnosing radiative forcing. In an attempt to refine our understanding of the ERF and its components, RFMIP2.0 will address two additional research questions:

3. What is the influence of the underlying climate state on radiative forcing?

4. To what extent is radiative forcing separable from radiative feedbacks when considering land processes?

As detailed in later sections, Question 3 is motivated by recent work that has explained how radiative forcing magnitude and its inter-model spread depend on the surrounding environment in which the perturbation occurs (Jeevanjee et al. 2021; He et al. 2023; Y.T. Chen et al. 2023). Question 4 is motivated by a longstanding inconsistency between formal



definitions of the ERF and the conventional approach of diagnosing ERF from fixed-SST experiments where land temperatures are allowed to change (Tang et al. 2019; Smith et al. 2020; Andrews et al. 2021)

We note that the CMIP6 iteration of RFMIP also included a component for benchmarking greenhouse gas IRFs in GCMs via a set of offline radiation calculations and a component for evaluating aerosol-radiation schemes by prescribing aerosol optical properties (Pincus et al. 2016, 2020). While these efforts continue to make valuable contributions to the development and evaluation of radiation schemes, not all of the relevant advancements in radiative transfer follow the CMIP timeline, so we are not proposing formal follow-up efforts here. However, many aspects of those protocols could be

implemented informally to evaluate individual CMIP7 models and we encourage interested modeling groups to contact the RFMIP2.0 leads for assistance.

In the remaining sections, we outline the requested experiments and model output that comprise the protocol of RFMIP2.0 and highlight the protocol's various synergies with other CMIP7 community Model Intercomparison Projects (MIPs). While we encourage modeling centers to perform the full protocol, we welcome partial participation and thus rank the

priority level of all requests into Tiers. Updates and clarifications to the RFMIP2.0 protocol will be maintained at https://rfmip.github.io/ throughout the project.

## 2 Experimental Protocol

### 2.1 Isolating the Radiative Forcing

In nature and in realistic model simulations, the radiative forcing and response typically happen simultaneously as perturbations to the atmosphere's composition continue to occur over time. Therefore, to isolate and diagnose the ERF and its components, the protocol for RFMIP2.0 centers around simulations that suppress the radiative response. Specifically, the protocol uses "fixed-SST" simulations, whereby an annual climatology of sea surface temperatures (SSTs) and sea ice concentrations (SICs) are prescribed identically in a control simulation and in perturbed forcing simulations (Hansen et al.

2005, Forster et al., 2016).  With no change in SSTs, and thus a suppressed surface-temperature-mediated radiative response (i.e. $\lambda \Delta T_s \approx 0$ in Eq. 1), the difference in top-of-atmosphere net radiation between a perturbed and control state serves as an estimate of the ERF (i.e. $N \approx F$ in Eq 1). In a strict sense this is just an approximation of ERF, however, since land temperatures are able to respond in standard fixed-SST simulations and thus radiative responses are not entirely eliminated. This is a motivation for the 4[th] research question.

Mimicking the CMIP6 iteration of RFMIP, in RFMIP2.0 the prescribed SSTs and SICs should come from a monthly-averaged climatology derived using at least 30 years of the same model's coupled pre-industrial control DECK simulation (*piControl)*, which is representative of 1850 conditions. The 30-year segment can come from any part of the *piControl* simulation, but the same derived SST and SIC climatologies should be used in all RFMIP2.0 experiments. When RFMIP2.0 experiments call for pre-industrial or historical configurations, the standard CMIP7 input4MIPs forcing datasets





(Dunne et al. 2025; Durack et al., 2025) should be used for the forcing agent boundary conditions (e.g. greenhouse gas concentrations and aerosol (precursor) emissions). All experiments should also include interactive vegetation if a model has this capability. For RFMIP2.0, we request two general types of fixed-SST simulations as detailed below: 30-year *timeslice* experiments and time-evolving *transient* experiments that span from 1850 to 2100.

**2.2 General Description of Timeslice and Transient Experiments**


The *timeslice* experiments are 30-year "fixed-SST" simulations with the pre-industrial SST and SIC annual climatologies prescribed repeatedly for each year of the integration. Likewise, forcing agent boundary conditions representative of a single year (or a multiplicative of a single year) are imposed repeatedly over the length of the simulation. In support of addressing Research Question 1, this approach is specifically designed to reduce noise and allow for a robust

diagnosis of a model's ERF to better than ± 0.05 W/m$^2$ (Forster et al. 2016). The timeslice control simulation, *piClim-control*, requires pre-industrial forcing boundary conditions representative of 1850, as per the *piControl* simulation. The accompanying set of *timeslice* perturbed simulations (Tables 1, 3 and 4), require present day forcing boundary conditions representative of 2021 for all or individual types of anthropogenic forcing agents (GHGs, aerosols, etc.), while others remain at pre-industrial levels. This allows for the diagnosis of the present-day ERF, and its key contributors, relative to pre-

industrial (similar to Figure 1). Present day is defined as 2021 because that will be the final year of the coupled *historical* and atmosphere-only *amip* CMIP7 DECK simulations. An exception to using present day forcing boundaries is the various $CO_2$ perturbation experiments described below, which instead require a multiple of 1850 $CO_2$ concentrations.

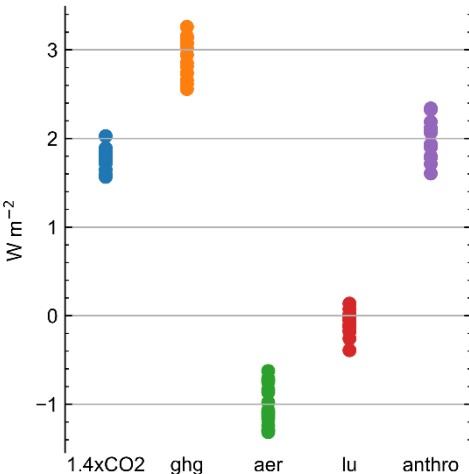

**Figure 1.** Global-mean Effective Radiative Forcing in 2014, relative to pre-industrial, for $CO_2$ (scaled from a 4x$CO_2$ simulation to approximately 1.4x$CO_2$ in 2014), well-mixed greenhouse gases (ghg), anthropogenic aerosols (aer), land use (lu), and all anthropogenic



forcings (anthro). Diagnosed in CMIP6 models using RFMIP experiments analogous to the Tier 1 time slice experiments described in Table 1.

The RFMIP2.0 protocol also calls for a set of *transient* experiments with forcing agents that vary over time from 1850 to 2100, designed to address Research Question 2. Individual transient experiments are detailed in Section 2.3 and Table 2 and are denoted by the "*piClim-hist*" prefix. In general, these experiments call for individual forcing agent types to be set to each respective year while all other forcing agents remain at 1850 pre-industrial conditions over the full length of the experiments. This "one-but-all" approach follows the protocol used in a set of fully-coupled, single forcing historical simulations from the

Detection and Attribution Model Intercomparison Project (DAMIP; Gillett et al. 2025), as described in Section 4.

While select forcing agents vary with time in the *transient* experiments, the SST and SIC fields should be fixed in time by repeatedly using the same climatologies prescribed in the *timeslice* experiments, for all years. Since radiative feedbacks are largely suppressed when SSTs/SICs are fixed, time-evolving ERFs relative to pre-industrial conditions can be diagnosed directly by differencing the timeseries of top-of-atmosphere radiative fluxes from a transient simulation and the time-averaged

radiative fluxes from the timeslice *piClim-control* simulation, which serves as the appropriate control. Analysis indicates that these transient estimates suffer from year-to-year noise (Pincus et al. 2016). In order to produce a more robust estimate of the transient ERF, the RFMIP2.0 protocol requests three ensemble members be performed for each transient experiment, if resources allow. Spanning from 1850 to 2100, the transient experiments require extending beyond the CMIP7 historical period that ends in 2021. Up to that year, the target forcing agent should be prescribed using the standard CMIP7 forcing input datasets

used in the fully-coupled historical simulations. From 2022–2100, the Medium concentration-driven scenario from ScenarioMIP should be used (*scen7-mc*), which is a scenario broadly consistent with current climate policies (Van Vuuren et al. 2025). In all years, greenhouse gases including $CO_2$ should be specified using prescribed concentrations (i.e. not in $CO_2$ emissions-driven mode for those models which have capability to do so).

The general experiment descriptions provided in this section apply across all experiments in the RFMIP2.0 protocol.

The next sections provide details on each specific experiment, organized by priority Tiers.

## 2.3 RFMIP Continuity and Relevant CMIP7 DECK and Fast Track Experiments

While RFMIP supports the fundamental scientific study of radiative forcing specifically, arguably its most important

contribution is in enabling a much wider breadth of scientific applications that require a systematic diagnosis of radiative forcing in models. RFMIP is as much a diagnostic service for the CMIP community as it is a scientific endeavor. Consequently, the highest priority Tier 1 simulations in the RFMIP2.0 protocol (Table 1) focus on continuity of the RFMIP experiments from CMIP6 that most directly address Research Questions 1 and 2 and that have the greatest applicability to the CMIP communities that RFMIP serve.


 

**Table 1:** Atmosphere-only timeslice experiments for diagnosing Effective Radiative Forcing and its components at present day and perturbed $CO_2$ conditions, from a pre-industrial control. All simulations include sea-surface temperatures and sea-ice concentrations fixed with a model-specific pre-industrial climatology, and include interactive vegetation when possible. The piClim-4xCO2-bgc experiment additionally requires a biogeochemical modeling component. A minimum of one ensemble member is requested. Experiments marked with * repeat experiments in the CMIP6 RFMIP protocol, for continuity.

| Experiment ID | Description | Years | Tier | CMIP7 DECK/Fast Track |
|---|---|---|---|---|
| piClim-control* | Pre-industrial (1850) condition baseline experiment | 30 | 1 | DECK |
| piClim-4xCO2* | CO2 concentrations set to 4 times pre-industrial | 30 | 1 | DECK |
| piClim-anthro* | Present-day (2021) anthropogenic forcing (greenhouse gases including ozone, aerosols, and land use) | 30 | 1 | DECK |
| piClim-aer* | Present-day anthropogenic aerosols | 30 | 1 | Fast Track |
| piClim-ghg* | Present-day well-mixed greenhouse gases only (non-ozone) | 30 | 1 | |
| piClim-lu* | Present-day land use | 30 | 1 | |
| piClim-0.5xCO2 | CO2 concentrations set to half of pre-industrial | 30 | 3 | |
| piClim-2xCO2 | CO2 concentrations set to 2 times pre-industrial | 30 | 3 | |
| piClim-4xCO2-bgc | CO2 concentrations set to 4 times pre-industrial applied only to carbon cycle | 30 | 3 | |
| piClim-4xCO2-rad | CO2 concentrations set to 4 times pre-industrial applied only to radiation | 30 | 3 | |

At the core of CMIP, the DECK (Diagnostic, Evaluation and Characterization of Klima) consists of well-established simulations used to quantify fundamental characteristics of a climate model. The DECK experiments are requested from all models and serve as an entryway into the larger CMIP environment (Eyring et al. 2016). Pointing to the importance of



diagnosing radiative forcing for model evaluation and a broad range of science topics, the CMIP7 DECK has been expanded
to include three timeslice experiments from the CMIP6 iteration of RFMIP: *piClim-control, piClim-4xCO2,* and *piClim-anthro*
(present-day anthropogenic forcings with all other boundary conditions set to pre-industrial). While these experiments are
technically not part of RFMIP2.0, we will treat them as such here and consider them to be Tier 1 RFMIP2.0 experiments. All
three are key to maximizing the scientific value of the RFMIP2.0 experiments described herein and as the control run for most
RFMIP2.0 experiments, performing *piClim-control* is essentially a requirement for RFMIP2.0 participation.

**Table 2:** Atmosphere-only transient experiments for diagnosing the time evolution of the Effective Radiative Forcing and its
components relative to the timeslice pre-industrial control experiment in Table 1. All simulations include sea-surface
temperatures and sea-ice concentrations fixed with a model-specific pre-industrial climatology, and include interactive
vegetation when possible. Forcing boundary conditions after 2021 come from the scen7-mc frozen policy, a medium pathway
scenario also used by DAMIP v2.0. A minimum of one ensemble member is requested, though three or more ensemble
members is preferred. Experiments marked with * repeat experiments in the CMIP6 RFMIP protocol for continuity.

| Experiment ID | Description | Start Year | End Year | Tier | CMIP7 DECK/Fast Track |
|---|---|---|---|---|---|
| piClim-histall* | Time-varying forcing conditions for all agents | 1850 | 2100 | 1 | Fast Track |
| piClim-histaer* | Time-varying anthropogenic aerosol forcing conditions | 1850 | 2100 | 1 | Fast Track |
| piClim-histnat* | Time-varying natural forcing conditions from volcanoes, solar irradiance (including spectral), variability, etc. | 1850 | 2100 | 2 | |
| piClim-histghg* | Time-varying well-mixed greenhouse gases only (non-ozone) | 1850 | 2100 | 2 | |


In addition to the DECK experiments, CMIP7 includes the "Fast Track"; a subset of experiments, selected from
community MIPs, that are particularly responsive to the needs of national and international assessments. To support these
efforts, the Fast Track experiments will typically be performed earlier than the rest of the CMIP7 protocol. Along with the
three new DECK experiments mentioned above, the Fast Track will include three additional Tier 1 experiments from
RFMIP2.0: a *timeslice*, fixed-SST experiment of present-day anthropogenic aerosol forcings with all other boundary conditions



set to pre-industrial (*piClim-aer*), an analogous *transient* anthropogenic aerosol forcing experiment (*piClim-histaer*) and a transient present-day all anthropogenic and natural forcings experiment (*piClim-histall*). Due to the accelerated timeline, some modeling centers may choose to use an earlier version of their GCM for the Fast Track and a later version for completing the

rest of the community MIP protocols. We welcome this approach for RFMIP2.0, but strongly encourage modeling centers to still perform the DECK and Fast Track experiments closely aligned with RFMIP2.0 using the newer model version. In particular, running the DECK *piClim-control* experiment, and running *piControl* to generate the model-specific SST and SIC climatologies, are required prerequisites for performing any other RFMIP2.0 experiment.

To complement and add value to the RFMIP-relevant DECK and Fast Track experiments, we include two additional

experiments under Tier 1: a *timeslice* experiment of present-day greenhouse gas forcings (*piClim-ghg*) and a *timeslice* experiment of present-day land use conditions (*piClim-lu*). Both were part of the CMIP6 RFMIP protocol and, with their inclusion, all major types of forcing agent perturbations (with the exception of ozone) incorporated into the *piClim-anthro* DECK experiment are included in Tier 1, allowing for a full decomposition of the total, present-day anthropogenic ERF.

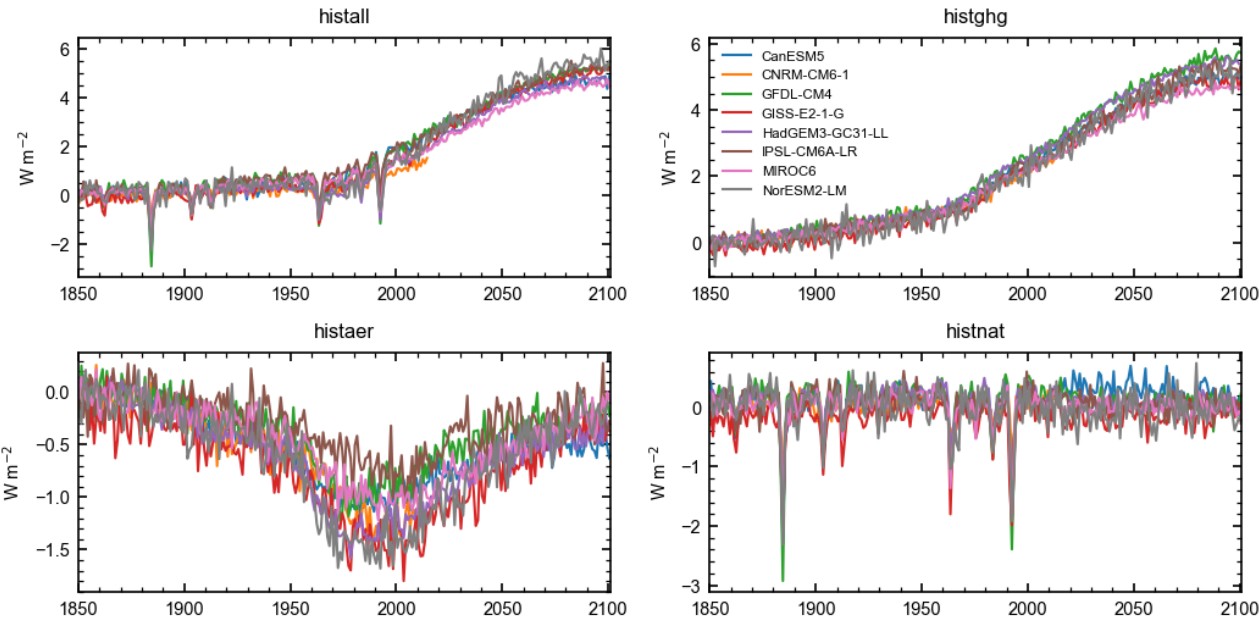

**Figure 2.** Global-mean timeseries of Effective Radiative Forcing from 1850 to 2100 (extended beyond 2014 using the SSP2-4.5 scenario) for all time-varying anthropogenic and natural forcings (histall) and, individually, for well-mixed greenhouse gases (histghg), anthropogenic aerosols (histaer) and natural forcings (histnat). Diagnosed in CMIP6 models using RFMIP experiments analogous to the RFMIP2.0 transient experiments tables described in Table 2.

The RFMIP2.0 protocol Tier 2 consists of two additional transient simulations spanning 1850 to 2100 that were also requested in the previous iteration of RFMIP: *piClim-histghg*, requiring time-evolving greenhouse gases and *piClim-histnat*, requiring time-evolving natural forcings from volcanoes and solar variability. These experiments, along with Tier 1 *piClim-*



*histaer*, can be used to decompose the total transient ERF diagnosed with Tier 1 *piClim-histall*, and will therefore be useful for attribution studies (also see Section 4). Likewise, the ERF from all four transient experiments (Figure 2) can be used to

isolate and decompose the role of forcing in the fully-coupled *historical* DECK simulations.

## 2.4 Clarifying the Definition of Radiative Forcing and its Sources of Spread

The expanding uses of the ERF metric have highlighted limitations in our understanding of radiative forcing and in the appropriateness of our diagnostic approaches. The remaining RFMIP2.0 experiments, considered Tier 3, focus on two emerging research themes that address the definition of radiative forcing and would benefit from a multi-model evaluation: the sensitivity of the ERF to the underlying base state (Question 3) and the proper accounting of surface changes when diagnosing the ERF (Question 4).

A change in radiation is determined not just by the size of the associated climate perturbation, but also by the characteristics of the base climate state in which the perturbation occurs. This radiative sensitivity to the climate state has been well-appreciated in the study of radiative feedbacks, as evident by the considerable, recent focus on the "pattern effect" of SST spatial distributions modulating feedbacks (Stevens et al. 2016; Andrews et al. 2022). Whether the ERF is sensitive to the underlying climate state has been comparatively less studied, but remains an important and timely question.

Recent work has clarified that the magnitude of the IRF for a given $CO_2$ concentration change is highly sensitive to base state temperatures, and particularly to the difference between temperatures in the stratosphere and at the surface (Huang et al. 2016; Jeevanjee et al. 2021; Romps et al. 2022). Since GCMs have different base states, this sensitivity contributes to the inter-model spread of the IRF (He et al. 2023; Byrom et al. 2025). Additionally, as the base state evolves, $CO_2$ concentration changes of the same magnitude but under future (or past) temperature states have a different IRF (He et al. 2023), which may

be relevant for interpreting paleoclimate or future, projected conditions that differ from the present. To understand the full implications of these processes on the climate response, it is important to confirm whether a similar base-state sensitivity extends to the ERF, as found in work by Mitevski et al. (2025). Doing so in a multi-model framework will not only ensure a robust assessment, but will also be useful for building representation of these sensitivities in climate model emulators, which usually represent forcing using simple expressions based solely on concentration changes (e.g. Etminan et al. 2016, Forster et

al. 2021).

    Motivated by this, the RFMIP2.0 protocol calls for the Tier 1 *piClim-4xCO2* experiment to be repeated, but with SSTs that are uniformly +4K warmer than the model's *piClim* SST climatology (*piClim-p4K-4xCO2*). Importantly, the timeslice control simulation must also be performed with the +4K warmer SSTs (*piClim-p4K*), since the intent is to evaluate the ERF under an alternate, but still fixed, SST state. Modeling centers are also encouraged to apply this +4K SST framework

to the present-day anthropogenic aerosol timeslice experiment *(piClim-p4K-aer)*. It is known that the aerosol IRF (direct effect) is sensitive to the base state (Stier et al. 2013) and that warmer SSTs will lead to a different cloud base state. Evidence suggests





these state dependencies therefore impact aerosol-cloud interactions and thus aerosol ERF (Lorian and Dagan 2024; S. Zhang et al. 2025) but this has not yet been evaluated systematically across models. We note that these +4K simulations (Table 3) do not test the sensitivity of the ERF to stratospheric base state temperatures, which is known to be important for $CO_2$ IRF

(Jeevanjee et al. 2021; He et al. 2023; Mitevski et al. 2025). Prescribing a stratospheric temperature may inhibit a model's ability to stratospherically adjust to the IRF in a realistic manner, which is a large component of the $CO_2$ ERF, so there is no obvious method to similarly test this sensitivity in a standardized, multi-model framework.

**Table 3:** Atmosphere-only timeslice experiments following analogous experiments in Table 1, but perturbed from a pre-
industrial control climate with 4K uniformly warmer sea surface temperatures to evaluate the sensitivity of the Effective Radiative Forcing to the underlying climate state. A minimum of one ensemble member is requested.

| Experiment ID | Description | Years | Tier | CMIP7 DECK/Fast Track |
|---|---|---|---|---|
| piClim-p4K | Pre-industrial (1850) conditions but with sea surface temperatures uniformly 4K warmer | 30 | 3 | Fast Track (for AerChemMIP) |
| piClim-p4K-4xCO2 | CO2 concentrations set to 4 times pre-industrial with sea surface temperatures uniformly 4K warmer than pre-industrial | 30 | 3 | |
| piClim-4K-aer | Present day (2021) aerosols with sea surface temperatures uniformly 4K warmer than pre-industrial | 30 | 3 | |

Also related to underlying climate conditions, recent work suggests there is an asymmetric warming versus cooling
response to an equal increase or decrease in $CO_2$ concentration, possibly being driven by an initial asymmetry in the ERF magnitude (Mitevski et al. 2022; Chalmers et al. 2022; Kay et al. 2024). To evaluate the robustness of this asymmetry across models and identify non-linearity in the relationship between ERF and concentration change, the Tier 3 set of experiments also includes *timeslice* perturbations with a doubling of $CO_2$ concentration (*piClim-2xCO2*) and a halving of $CO_2$ concentration (*piClim-0.5xCO2*) from pre-industrial (Table 1). As in all other RFMIP experiments, interactive vegetation should be enabled
if a model includes that capability. These experiments will complement fully-coupled doubling and halving experiments (*abrupt-2xCO2* and *abrupt-0.5xCO2*) included in the CMIP7 Fast Track. Along with the respective quadrupling $CO_2$



experiments, this will allow for a complete diagnosis of the asymmetry in the temperature response to $CO_2$ perturbations and the contribution to this from forcing.

The remaining Tier 3 experiments are designed to study a long-maintained discrepancy between the definition of ERF and how it is traditionally diagnosed. In nearly all cases, including in CMIP and all experiments described here so far, land surface temperature (*LST*) is free to evolve in fixed-SST simulations used to diagnose the ERF. Since the ERF equals the total radiative change from these experiments, it therefore includes any radiative responses to the ∆*LST*, thus often diverging from the formal definition of ERF, which excludes radiative responses to a change in global-mean surface temperature (Forster et al. 2021). This discrepancy has been accepted by the community because fixing *LST* has been considered technically challenging in GCMs. However, two recent successful attempts by Andrews et al. (2021) and Zhang et al. (2025) show that prescribing *LST* along with SSTs for this purpose is not only possible in current GCMs, but impactful. Both studies find the effects of land warming reduce the estimates of 4xCO2 ERF by ~15%, implying a proportional bias in any ECS estimate using the ERF. And while there are multiple proposed workarounds to remove ∆*LST* effects from the ERF in standard fixed-SST simulations after-the-fact, they each miss key components of the total effect that are difficult to isolate without a dedicated experiment, or are not yet well-understood enough to be represented by a simplified correction (Hansen et al. 2005; Tang et al. 2019; Smith et al. 2020; Andrews et al. 2021).

In an attempt to further understand the effects of ∆*LST* on ERF, investigate potential correction methods, and directly diagnose an ERF in a manner closer to its formal definition, we request modeling centers repeat all Tier 1 *timeslice* experiments, but with fixed land temperatures in addition to fixed-SSTs as described in Table 4, including the control experiment (*piClim-FixedLST*). This will be the first coordinated, multi-model effort to perform fixed-*LST*, fixed-SST 4xCO2 experiments and the first attempt at all to perform these experiments for non-$CO_2$ perturbations. Given the technical challenges of this exercise, likely the precise method for fixing land temperatures will differ between models. Therefore, we provide some general guidelines here, but acknowledge the typical expectation of stringent standardization across models will need to be relaxed for these experiments. Accordingly, we caution simulations users against over-interpreting inter-model spread in these *FixedLST* experiments as a measure of physical uncertainty, since implementation techniques may differ across models more than usual for these experiments.

Following Ackerley and Dommenget (2016), Andrews et al. (2021), and B. Zhang et al. (2025), 3-hourly output of land surface temperature should be saved from all years of a standard, 30-yr Tier 1 *piClim-control* simulation with freely evolving land conditions. This output is then used to prescribe the surface temperature at pre-industrial levels in all *FixedLST* 30-yr experiments, which are otherwise a rerun of the standard *timeslice* experiments. Specifically, this 3-hourly output should come from the *piClim-control* simulation that is being used as the control for all Tier 1 and Tier 2 RFMIP2.0 experiments. For many models, this will be the simulation that was submitted to the CMIP7 DECK. However, if 3-hourly data was not previously saved for that particular simulation, it is acceptable to perform another ensemble member of *piClim-control* to generate the higher frequency output. Prescribing high temporal resolution data in the *FixedLST* experiments is necessary in order to preserve the land surface diurnal cycle. Importantly, to avoid nonlinear effects and provide a more regulated assessment, the



control for the perturbed-forcing *FixedLST* simulations should likewise be a pre-industrial forced simulation that mimics *piClim-control*, but with prescribed LST (*piClim-FixedLST*). The *piClim-FixedLST* and the standard *piClim-control* simulations should have similar surface, atmospheric, and radiative climatologies. Verifying this during the development process is an important step towards determining whether the *FixedLST* simulations include realistic responses and whether

they will serve as adequate counterparts to the free-land *timeslice* simulations, in an effort to accurately isolate land effects on the ERF. Work by Ackerley et al. (2018) provides a framework for conducting this evaluation of the climatologies.

**Table 4:** Atmosphere-only timeslice experiments following analogous experiments in Table 1, but with land surface temperatures additionally fixed (FixedLST), along with sea-surface temperatures and sea-ice concentrations. To preserve the
335 diurnal cycle, land temperatures are prescribed with model-specific, 3-hourly data from the fixed-SST (with freely evolving land) control experiment in Table 1 (piClim-control). Soil moisture and soil temperatures may also be similarly prescribed, particularly if a given land model requires this in order to maintain fixed surface temperatures. A minimum of one ensemble member is requested.

| Experiment ID | Description | Years | Tier |
|---|---|---|---|
| piClim-FixedLST | Pre-industrial (1850) conditions with land-surface temperatures fixed from piClim-control | 30 | 3 |
| piClim-FixedLST-4xCO2 | CO2 concentrations set to 4 times pre-industrial with land surface temperatures fixed at pre-industrial | 30 | 3 |
| piClim-FixedLST-anthro | Present-day (2021) anthropogenic forcing (greenhouse gases, aerosols, and land use) with land surface temperatures fixed at pre-industrial | 30 | 3 |
| piClim-FixedLST-aer | Present-day anthropogenic aerosols with land surface temperatures fixed at pre-industrial | 30 | 3 |
| piClim-FixedLST-ghg | Present Day well-mixed greenhouse gases only (non-ozone) with land surface temperatures fixed at pre-industrial | 30 | 3 |
| piClim-FixedLST-lu | Present-day land use with land surface temperatures fixed at pre-industrial | 30 | 3 |

Although not specifically required in this protocol, in practice, for some models it may be necessary to save and prescribe soil temperature and soil moisture at multiple layers in order to fix surface temperatures and avoid unrealistic surface




moisture and hydrological responses, as in Ackerley and Dommenget (2016) and B. Zhang et al. (2025). Likewise, in some models, prescribing fixed-LST in a general sense may require prescribing multiple surface temperature variables, such as radiative surface temperature and vegetation canopy surface temperature (B. Zhang et al. 2025). However, these prescriptions should only be made when necessary and should be documented in the output netCDF metadata, in appropriate CMIP7 model documentation venues, or should be reported to the authors of this article so model-specific prescriptions can be noted on the RFMIP website.

It remains an open question whether all land temperatures need to be fixed in order to adhere to the strict definition of the ERF. Some of the $\Delta LST$ stems from local processes not directly induced by a radiation imbalance or by its classical global-mean temperature response. Most prominently, the vegetation physiological response to $CO_2$ increase causes land warming as plants close their stomata, reducing water release, and reducing evaporative cooling (Field et al 1995; Doutriaux-Boucher et al. 2009). Andrews et al. (2021) found warming from vegetation physiological changes accounted for roughly half of the $\Delta LST$ effect on the ERF for $4xCO_2$. Since this process is a local, direct adjustment to the $CO_2$ increase, any associated warming could arguably be classified as part of the forcing (Quaas et al. 2024). To better quantify the plant physiological effects on the ERF, and bound its uncertainty, the Tier 3 experiments consist of additional 30-yr timeslice experiments where a $4xCO_2$ perturbation is seen only by the radiation scheme, while the biogeochemical cycle model component, including the vegetation schemes, only see pre-industrial, $1xCO2$ concentrations (*piClim-4xCO2-rad*). For completeness, we also request the converse experiment, where the radiation scheme sees $1xCO2$ and the biogeochemical components see $4xCO2$ (*piClim-4xCO2-bgc*). When estimating the ERF, both should be paired with the Tier 1 *piClim-control* experiment as the control run. These *bgc* and *rad* experiments should have fixed-SSTs but freely-evolving *LST* so the effects of plant physiology-induced land warming on the ERF can be captured when compared to *piClim-4xCO2-FixedTs* and *piClim-4xCO2*. Such an assessment will be useful for a variety of carbon cycle applications outside of the scope of RFMIP.

The RFMIP2.0 experiment protocol outlined here balances a need for continuity with a need for scientific advancement, as the quantification of radiative forcing increasingly becomes part of the fabric of CMIP activities and model evaluation more generally. Since RFMIP-like experiments are part of the CMIP7 DECK, the experiments proposed here largely follow, and build from, those experiments in an effort to streamline participation from modeling centers and add value to both RFMIP2.0 and the DECK/Fast Track experiments.

## 3 Variable Data Request

The RFMIP2.0 model data output request includes the minimum variables needed to diagnose the ERF and its IRF and radiative adjustment components globally, using common methods like the radiative kernel technique (Soden et al. 2008; Smith et al. 2020). For reference, the request presented here includes all of the variables found in the "Diagnosing Radiative Forcing" Opportunity included in the CMIP7 Fast Track data request, plus some additional variables found in the "Baseline Climate Variables for Earth System Modeling" and the "Clouds, Circulation and Climate Sensitivity: Baseline" Opportunities



(Dingley et al. 2025; Jukes et al. 2025). Only monthly output is being requested for public dissemination, but as detailed in Section 2.5, some 3-hourly variable output from *piClim-control* will need to be generated in order to perform the *FixedLST* experiments. Given the reliance on monthly-mean data only, the burden on modeling centers to fulfill this data request is relatively low, but we still split the request into high priority Tier A variables and lower priority Tier B variables for any participant who is particularly resource limited. These tiers are applicable for the output of any RFMIP experiment. Table A1 in the Appendix outlines the variable data request, but we also provide a general description in this Section.

       The core application of RFMIP experiments is to diagnose the ERF of a model. Therefore, Tier A of the variable request includes all all-sky and clear-sky, longwave (LW) and shortwave (SW) radiative flux variables at the top-of-atmosphere and surface (e.g. *rlut* and *rlus*). The fluxes at both boundaries can additionally be differenced from each other to diagnose the ERF component of atmospheric radiative cooling, which drives the fast global precipitation response to a forcing perturbation (Myhre et al. 2018). In studies of the atmospheric radiative constraint on precipitation, it is necessary to consider turbulent fluxes, so latent and sensible heat fluxes are included in the data request as well, along with precipitation itself. Additionally, the Tier A energetic flux request includes vertically resolved upwelling and downwelling radiative flux profiles (e.g. *rlu*) and air temperature tendencies. A growing body of work has shown the benefits of analyzing vertical changes in radiation to build a physical understanding of the ERF and its components (Salvi et al. 2021; Allen et al. 2023). Tier A also includes a variety of non-energetic, standard 3D and 4D atmospheric variables needed for diagnosing rapid radiative adjustments, such as standard cloud properties, specific humidity profiles and temperature profiles.

       The Tier B variables are of near-equal value to those in Tier A, but are specialized and may require additional effort or computational expense to produce them. This includes a set of output from so-called "double-call" radiative transfer calculations; the most direct and accurate method to diagnose the *IRF* in a model (Collins et al. 2006; Chung and Soden 2015a). In this setup, for each radiation time step of a simulation, a second, offline call is made to the radiation scheme with identical environmental inputs as the first, online call, except for an alternative concentration in a given forcing agent. This is akin to the second radiation calls performed with no clouds to compute clear-sky conditions in GCMs, which have no impact on subsequent time steps. Following this approach, we request all-sky and clear-sky radiative flux diagnostics from additional, offline radiation calls with $CO_2$ concentration quadrupled (e.g. rlu4co2 and rlucs4co2) relative to the initial radiation call. The *IRF* for 4xCO2 can then be diagnosed by subtracting the corresponding fluxes from the original, online calls (requested in Tier A) from these "double-call" fluxes. Groups should prioritize providing these 4co2 fluxes from the *piClim-control* simulation, but providing them from the *piClim-p4K* experiment would also be highly beneficial as a means for assessing the sensitivity of the IRF to the underlying base climate state. In general, community demand for these *4co2* fluxes has greatly increased since CMIP6 (e.g. Soden et al. 2018; Fiedler et al. 2024), given a growing body of work highlighting the significant amount of $CO_2$ *IRF* spread that remains and open questions about its causes and its direct contribution to spread in subsequent radiative adjustments (e.g. He et al. 2023). Furthermore, double-calls provide the only direct method of diagnosing the IRF in a model, and using indirect methods like the radiative kernel technique to diagnose $CO_2$ IRF in particular has proven susceptible to biases (Chung and Soden 2015a; Kramer et al. 2019; Smith et al. 2021). Like a budget closure analysis, the difference between





the true IRF from double-calls and a kernel-derived estimate of the IRF, can serve as a measure of bias in the sum of kernel-derived estimates of radiative feedbacks and adjustments as well (Chung and Soden 2015b, Myhre et al. 2018).

Radiative fluxes from additional, offline radiation calls with pristine, no aerosol conditions are similarly requested under all-sky (e.g. rlutaf) and clear-sky (e.g. rlutcsaf) conditions, as noted in Table A1. These second calls should be performed for the *piClim-control, piClim-aer* and *piClim-histaer* simulations. Again, providing them for the *piClim-p4K* and *piClim-p4K-aer* experiments as well would be very beneficial for evaluating base state sensitivities. Following Ghan (2013), through a specific differencing of these fluxes with their respective initial radiation calls, and between experiments, one can directly decompose the present-day aerosol ERF into its aerosol-radiation interaction (ERFari) and aerosol-cloud interaction (ERFaci) components. These double-calls can also be used to diagnose the aerosol IRF and to validate the more detailed but indirect Approximate Partial Radiative Perturbation decomposition (Taylor et al. 2007), once cloud masking effects are properly accounted for (Zelinka, Smith et al. 2023, see their Appendix).

Diagnosing cloud radiative adjustments have become a core function of RFMIP, given their diverse contribution to the magnitude and uncertainty of ERF across forcing agents, and given their difficulty to constrain (Smith et al. 2018, 2020; Bellouin et al. 2020). Standard, Tier A variables can be used to diagnose cloud radiative adjustments in a cloud-mean, bulk sense for all forcing scenarios (Soden et al. 2004; Chung and Soden 2015b; Smith et al. 2018), and with some additional decomposition possible for aerosol ERF (Taylor et al. 2007; Zelinka, Smith et al. 2023). However, none of these methods provide granular, isolated information about the contribution of different cloud types (or types of aerosol-cloud interactions) that is necessary to understand cloud adjustments at a process level. To enable a more detailed decomposition of cloud radiative adjustments, we request a set of Tier B variables that require performing simulations with the COSP satellite simulator package turned on (Bodas-Salcedo et al. 2011; Swales et al. 2018). First, the *clisccp* variable is requested for all RFMIP experiments, which provides joint histograms of cloud fraction binned by cloud optical depth (tau) and cloud top pressure (CTP), and is produced from the International Satellite Cloud Climatology Project (ISCCP) satellite instrument simulator (Klein and Jakob 1999; Webb et al. 2001). This increasingly common output (provided by a majority of participating models in RFMIP for CMIP6) can be multiplied by cloud radiative kernels to isolate the contribution of changes in cloud amount, altitude and optical depth to total cloud radiative adjustment (Zelinka et al. 2012; Lee and Oreopoulos 2025).

Additionally, the Tier B request includes a set of cloud variables from the Moderate Resolution Imaging Spectroradiometer (MODIS) satellite instrument simulator (Pincus et al. 2012), including new joint histogram diagnostics that partition cloud fraction by cloud liquid water path (or ice water path) and cloud drop effective radius (*clmodis_lwpr* and *clmodis_iwpr*). These diagnostics have been implemented in the latest version of COSPv2.0 (CFMIP, 2025) and are analogously available in observations (Pincus et al. 2022). They enable the decomposition of ERFaci into contributions from Twomey effect, water path adjustments, and cloud fraction adjustments following Duran et al. (2025). Recognizing that fewer modeling centers have implemented the MODIS simulator, we stress that these MODIS diagnostics (and all other Tier B variable requests) are encouraged but strictly optional for participation in RFMIP2.0.



445

## 4 Synergy with other CMIP7 Community MIPs

450

Since diagnosing radiative forcing is a building block for many climate research applications, this RFMIP2.0 protocol complements numerous other CMIP7 Community MIPs linked scientifically to the study of climate sensitivity or climate-radiation interactions more broadly. For instance, knowing the radiative forcing of a GCM is key to understanding model responses to climate intervention techniques, as studied in GeoMIP (Visioni et al. 2023), or the responses to carbon cycle management – the focus of CDRMIP (Keller et al. 2018), relevant to C4MIP (Jones et al. 2016; Sanderson et al. 2024), and a driving research question for CMIP7 overall (Dunne et al. 2025). It is also valuable for exploring tipping points and the potential irreversibility of some climate impacts, which is the focus of TIPMIP (Winkelmann et al. 2025) and another driving research question for CMIP7 overall. Quantifying radiative forcing model diversity will be important for attributing observed increases in Earth's energy imbalance and addressing known model-observation trend discrepancies (Hodnebrog et al. 2024; Myhre et al. 2025). This is the focus of CERESMIP (Schmidt et al. 2023), which recommends performing *piClim-histall* in its protocol.

Additionally, a selection of community MIPs have a more specific connection to RFMIP2.0, through intentional overlap of protocols. Although not required, we encourage modeling centers participating in RFMIP2.0 to perform these corresponding, non-RFMIP2.0 experiments described below.

As noted in Section 2.5, the proposed *piClim-2xCO2* and *piClim-0.5xCO2* RFMIP2.0 Tier 3 experiments are ideally paired with *abrupt-2xCO2* and *abrupt-0.5xCO2,* which are included in the Fast Track via the upcoming CMIP7 iteration of CFMIP. These pairings will enable an accurate estimate of Equilibrium Climate Sensitivity and an evaluation of any nonlinear responses to $CO_2$ perturbation size. This evaluation will be important for interpreting variability in historically-forced and projection experiments where the magnitude of $CO_2$ perturbations change over time, such as those from ScenarioMIP (van Vurren et al. 2025), the Large Ensemble Single Forcing (LESF) MIP (D.M. Smith et al. 2022) and the CMIP7 iteration of DAMIP (v2.0; Gillett et al. 2025).

As in the respective CMIP6 iterations of the projects, RFMIP2.0 and DAMIP v2.0 include coordinated pairs of fixed-SST and fully-coupled *transient* experiments that follow the "one-but-all" methodology of perturbing one forcing type while keeping all others at pre-industrial levels. These proposed experiments all cover the *historical* DECK experiment time period, and extend beyond it using the ScenarioMIP "Medium" forcing scenario conditions run in concentration driven mode (*scen7-mc*). Together, these experiments will enable a more detailed analysis of historical changes, allowing users to attribute and decompose the historical variability into contributions from time-evolving radiative forcing versus climate response from 1850 to 2035 (the proposed end year of the DAMIP v2.0 experiments). Specifically, to conduct this decomposition for the total



variability, the RFMIP2.0 *piClim-histall* experiment should be paired with the *historical* experiment extended with the *scen7-mc,* which are included in the CMIP7 DECK and Fast Track, respectively. To further attribute by forcing perturbation type, the RFMIP2.0 *piClim-histaer, piClim-histghg,* and *piClim-histnat* should be paired with the DAMIP v2.0 fully-coupled *hist-aer*, *hist-ghg,* and *hist-nat*, respectively. This type of analysis will also be broadly relevant in CFMIP efforts to quantify uncertainty in the radiative response.

With the shared goal of quantifying the impacts of changing atmospheric composition on the climate, RFMIP2.0 and the CMIP7 iteration of AerChemMIP (AerChemMIP-2) complement each other on multiple fronts. Using an experimental design identical to the RFMIP2.0 Tier 1 *timeslice* experiments, AerChemMIP-2 calls for a suite of *piClim-X* experiments where $X$ is an individual forcer (e.g. $SO_2$, $CH_4$, $O_3$ and others). These experiments will allow users to further decompose the broader ERFs from RFMIP2.0 into contributions from individual constituents while, in turn, the RFMIP2.0 experiments will help put the single-constituent ERFs from AerChemMIP-2 into a wider context among forcing types. Note that both projects require performing *piClim-control* following the CMIP7 DECK definition.

AerChemMIP-2 also consists of a set of transient *histSST-pi* experiments which, like the RFMIP2.0 *piClim-hist* set of experiments, enable diagnosis of the time-evolving historical radiative forcing, including from individual aerosol constituents and reactive gases. Importantly, the experimental designs differ from the comparable RFMIP2.0 simulations, and this can be exploited. In contrast to the "one-but-all" RFMIP2.0 approach, the *histSST-piX* experiments use historical conditions (including time-evolving SSTs) as the reference state and perturb a given species back to pre-industrial conditions to diagnose the ERF ("all-but-one" approach). Pairing the two types of experiments can thus be used to assess nonlinear responses of the forcing to the underlying meteorological and compositional state – a topic that has been largely limited to single-model analysis (Simpson et al. 2023; S. Zhang et al. 2025). For a direct comparison, RFMIP2.0 and AerChemMIP-2 both call for a transient aerosol forcing experiment (*piClim-histaer* and *histSST-piAer*, respectively) which has previously been exploited to boost the number of models from which transient historical aerosol forcing is available (Smith et al. 2021a).

### 5 Summary

The Radiative Forcing Model Intercomparison Project for CMIP7 (RFMIP2.0) addresses a need for the systematic characterization of radiative forcing in global models. Through a set of timeslice and transient atmosphere-only simulations with fixed-SSTs, RFMIP2.0 will help determine the past and future Effective Radiative Forcings (ERFs) from greenhouse gases, aerosols, land use and natural sources, thus serving as a foundation for projecting the climate response to human and natural activity. With the inclusion of RFMIP experiments in the CMIP7 DECK, including *piClim-4xCO2*, it will become standard practice to use RFMIP-based estimates of ERF in evaluations of climate sensitivity and its uncertainty. Performing the RFMIP2.0 protocol will therefore allow for an expanded interpretation of a model's climate sensitivity and its projections of future climate, relative to other CMIP7 models. Since RFMIP2.0 carries over many experiments from the CMIP6 iteration



of the project, participation will also enable an evaluation of ERF across model generations, which is informative for model development.

Through newly introduced experiments, the RFMIP2.0 protocol also aims to improve scientific understanding of radiative forcing and address open questions about the definition of ERF. Namely, a set of experiments with uniformly perturbed SSTs will test the sensitivity of the ERF to the underlying climate state while a set experiments with fixed land temperatures will allow for arguably a more definitionally appropriate estimate of the ERF and its uncertainty by quantifying and evaluating the radiative effects of land temperature changes.

Since quantifying radiative forcing is fundamental to many climate applications, RFMIP2.0 will enhance a broad range of CMIP7 activities. With specific overlaps with the AerChemMIP2, DAMIP2 and CFMIP protocols, using select experiments from these other Community MIPs in tandem with RFMIP2.0 experiments will allow for a more granular evaluation of present-day ERF and its role as a driver of historical and future climate variability.

**Appendix**

**Table A1:** Variable output data request for RFMIP7 ranked by priority Tier A and secondary Tier B. All variables are requested at monthly temporal resolution for the full length of the simulations. All variables should be produced for all experiments except those marked with an * which are only necessary for piClim-control and piClim-p4K or those marked with a ^ which are only necessary for piClim-control, piClim-aer, their p4K equivalents, and piClim-histaer.

| Variable Short Name | Description | Table ID | Units | #Dims | Tier |
|---|---|---|---|---|---|
| **Energy and Precipitation** | | | | | |
| rsdt | TOA Incident Shortwave Radiation | Amon | W m$^{-2}$ | 3 | A |
| rsut | TOA Outgoing Shortwave Radiation | Amon | W m$^{-2}$ | 3 | A |
| rsutcs | TOA Outgoing Clear-Sky Shortwave Radiation | Amon | W m$^{-2}$ | 3 | A |
| rlut | TOA Outgoing Longwave Radiation | Amon | W m$^{-2}$ | 3 | A |
| rlutcs | TOA Outgoing Clear-Sky Longwave Radiation | Amon | W m$^{-2}$ | 3 | A |
| rsds | Surface Downwelling Shortwave Radiation | Amon | W m$^{-2}$ | 3 | A |
| rsdscs | Surface Downwelling Clear-Sky Shortwave Radiation | Amon | W m$^{-2}$ | 3 | A |





| rsus | Surface Upwelling Shortwave Radiation | Amon | W m$^{-2}$ | 3 | A |
|------|----------------------------------------|------|-----------|---|---|
| rsuscs | Surface Upwelling Clear-Sky Shortwave Radiation | Amon | W m$^{-2}$ | 3 | A |
| rlds | Surface Downwelling Longwave Radiation | Amon | W m$^{-2}$ | 3 | A |
| rldscs | Surface Downwelling Clear-Sky Longwave Radiation | Amon | W m$^{-2}$ | 3 | A |
| rlus | Surface Upwelling Longwave Radiation | Amon | W m$^{-2}$ | 3 | A |
| rluscs | Surface Upwelling Clear-Sky Longwave Radiation [identical to rlus in some models] | Amon | W m$^{-2}$ | 3 | A |
| rld | Downwelling Longwave Radiation | CFmon | W m$^{-2}$ | 4 | A |
| rldcs | Downwelling Clear-Sky Longwave Radiation | CFmon | W m$^{-2}$ | 4 | A |
| rlu | Upwelling Longwave Radiation | CFmon | W m$^{-2}$ | 4 | A |
| rlucs | Upwelling Clear-Sky Longwave Radiation | CFmon | W m$^{-2}$ | 4 | A |
| rsd | Downwelling Shortwave Radiation | CFmon | W m$^{-2}$ | 4 | A |
| rsdcs | Downwelling Clear-Sky Shortwave Radiation | CFmon | W m$^{-2}$ | 4 | A |
| rsu | Upwelling Shortwave Radiation | CFmon | W m$^{-2}$ | 4 | A |
| rsucs | Upwelling Clear-Sky Shortwave Radiation | CFmon | W m$^{-2}$ | 4 | A |
| tnt | Tendency of Air Temperature | CFmon | K s-1 | 4 | A |
| hfls | Surface Upward Latent Heat Flux | Amon | W m$^{-2}$ | 3 | A |
| hfss | Surface Upward Sensible Heat Flux | Amon | W m$^{-2}$ | 3 | A |
| pr | Precipitation | Amon | kg m$^{-2}$ s$^{-1}$ | 3 | A |





| od550aer^ | Ambient Aerosol Optical Thickness at 550nm | AERmon | 1 | 3 | B |
|---|---|---|---|---|---|
| rsut4co2* | TOA Outgoing Shortwave Radiation in 4XCO2 Atmosphere | CFmon | W m$^{-2}$ | 3 | B |
| rsutcs4co2* | TOA Outgoing Clear-Sky Shortwave Radiation 4XCO2 Atmosphere | CFmon | W m$^{-2}$ | 3 | B |
| rlut4co2* | TOA Outgoing Longwave Radiation 4XCO2 Atmosphere | CFmon | W m$^{-2}$ | 3 | B |
| rlutcs4co2* | TOA Outgoing Clear-Sky Longwave Radiation 4XCO2 Atmosphere | CFmon | W m$^{-2}$ | 3 | B |
| rld4co2* | Downwelling Longwave Radiation 4XCO2 Atmosphere | CFmon | W m$^{-2}$ | 4 | B |
| rldcs4co2* | Downwelling Clear-Sky Longwave Radiation 4XCO2 Atmosphere | CFmon | W m$^{-2}$ | 4 | B |
| rlu4co2* | Upwelling Longwave Radiation 4XCO2 Atmosphere | CFmon | W m$^{-2}$ | 4 | B |
| rlucs4co2* | Upwelling Clear-Sky Longwave Radiation 4XCO2 Atmosphere | CFmon | W m$^{-2}$ | 4 | B |
| rsd4co2* | Downwelling Shortwave Radiation 4XCO2 Atmosphere | CFmon | W m$^{-2}$ | 4 | B |
| rsdcs4co2* | Downwelling Clear-Sky Shortwave Radiation 4XCO2 Atmosphere | CFmon | W m$^{-2}$ | 4 | B |
| rsu4co2* | Upwelling Shortwave Radiation 4XCO2 Atmosphere | CFmon | W m$^{-2}$ | 4 | B |
| rsucs4co2* | Upwelling Clear-Sky Shortwave Radiation 4XCO2 Atmosphere | CFmon | W m$^{-2}$ | 4 | B |
| rlutaf^ | TOA Outgoing Aerosol-Free Longwave Radiation | AERmon | W m$^{-2}$ | 3 | B |
| rlutcsaf^ | TOA Outgoing Clear-Sky, Aerosol-Free Longwave Radiation | AERmon | W m$^{-2}$ | 3 | B |



| rsutaf^ | TOA Outgoing Aerosol-Free Shortwave Radiation | AERmon | W m$^{-2}$ | 3 | B |
|---|---|---|---|---|---|
| rsutcsaf^ | TOA Outgoing Clear-Sky, Aerosol-Free Shortwave Radiation | AERmon | W m$^{-2}$ | 3 | B |
| tntrl | Tendency of Air Temperature due to Longwave Radiative Heating | Emon | K s-1 | 4 | B |
| tntrlcs | Tendency of Air Temperature due to Clear Sky Longwave Radiative Heating | Emon | K s-1 | 4 | B |
| tntrs | Tendency of Air Temperature due to Shortwave Radiative Heating | Emon | K s-1 | 4 | B |
| tntrscs | Tendency of Air Temperature due to Clear Sky Shortwave Radiative Heating | Emon | K s-1 | 4 | B |
| **Thermodynamic** | | | | | |
| ps | Surface Air Pressure | Amon | Pa | 3 | A |
| ts | Surface (Skin) Temperature | Amon | K | 3 | A |
| tas | Near-Surface Air Temperature | Amon | K | 3 | A |
| ta | Air Temperature | Amon | K | 4 | A |
| hur | Relative Humidity | Amon | % | 4 | A |
| hus | Specific Humidity | Amon | 1 | 4 | A |
| zg | Geopotential Height | Amon | m | 4 | A |
| **Clouds** | | | | | |
| clt | Total Cloud Cover Percentage | Amon | % | 3 | A |
| clwvi | Condensed Water Path | Amon | kg m$^{-2}$ | 3 | A |
| clivi | Ice Water Path | Amon | kg m$^{-2}$ | 3 | A |
| cl | Cloud Cover Percentage | Amon | % | 4 | A |
| clw | Mass Fraction of Cloud Liquid Water | Amon | kg kg$^{-1}$ | 4 | A |
| cli | Mass Fraction of Cloud Ice | Amon | kg kg$^{-1}$ | 4 | A |
| clisccp | ISCCP Cloud Area Percentage | CFmon | % | 5 | B |



| clmodis | MODIS Cloud Area Fraction | CFmon | % | | B |
|---|---|---|---|---|---|
| clmodisliquid | MODIS Liquid-Topped Cloud Area Fraction | CFmon | % | | B |
| clmodisice | MODIS Ice-Topped Cloud Area Fraction | CFmon | % | | B |
| clmodis_lwpr (clmodisliquidReff) | MODIS Liquid-Topped Cloud Area Fraction reported on droplet effective radius dimensions | CFmon | % | | B |
| clmodis_iwpr (clmodisiceReff) | MODIS Ice-Topped Cloud Area Fraction reported on droplet effective radius dimensions | CFmon | % | | B |

**Author Contributions**

As co-leads of RFMIP2.0, all authors (RJK, CS, TA) contributed to the design of the experiment protocols described in this work.  RJK led writing of the manuscript and CS developed the figures.  All authors contributed to revisions of the text.

**Competing Interests**

        The authors declare they have no competing interests or conflicts of interest.

**Code and Data Availability**

        Model output from simulations described here as part of the RFMIP2.0 protocol will be distributed through the Earth System Grid Federation (ESGF), with associated metadata and documentation following standard CMIP7 formats. Updates to the proposed protocol and additional information about the model output will be provided at https://rfmip.github.io/.  Output

data from CMIP6 used to produce Figure 1 and Figure 2 is similarly publicly available from the ESGF. Code used to generate those figures is available on Zenodo at https://doi.org/10.5281/zenodo.17087560 (C. Smith, 2025).



## Acknowledgments

We thank Drs. David Paynter, Vaishali Naik, and Robert Pincus for thoughtful and valuable reviews of this
manuscript and we thank many others in the extended RFMIP community for fruitful discussions and groundbreaking work
that informed this protocol. We also want to express our gratitude to Drs. Robert Pincus, Piers Forster, and Bjorn Stevens for
pioneering the first iteration of RFMIP and entrusting us to build from it. R.J.K. acknowledges support from the Geophysical
Fluid Dynamics Laboratory (GFDL), NOAA, but the statements, findings, conclusions and recommendations are those of the
author(s) and do not necessarily reflect the views of the National Oceanic and Atmospheric Administration or the US.
Department of Commerce. C.S. was supported by the Horizon Europe research and innovation programs under grant
agreements 101081661 (WorldTrans) and 101081369 (SPARCCLE). T.A. was supported by the Met Office Hadley Centre
Climate Programme funded by DSIT.

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
