# Peer review of "The Radiative Forcing Model Intercomparison Project (RFMIP2.0) for CMIP7"

_EGUsphere, 2025_

## Referee Comment (RC1)

Review of "The Radiative Forcing Model Intercomparison Project (RFMIP2.0) for CMIP7" by Kramer et al. egusphere-2025-4378

**Summary**

The authors present the experimental design and data request for the next iteration of RFMIP. The protocol is designed to answer four main motivating questions regarding the magnitude and components of present-day radiative forcing, how forcing components have evolved historically and will evolve into the future, how climate state influences forcing, and how land responses affect the diagnosed radiative forcing. Several of the proposed experiments follow from the CMIP6 iteration of RFMIP and will provide a consistent point of comparison with previous analyses; several new experiments are requested to address emerging questions; and several previous RFMIP experiments are no longer part of the official protocol (but are instead encouraged to continue informally).

RFMIP serves as a crucial piece of CMIP, providing estimates of radiative forcing that are needed for a large number of scientific studies, including many done as part of other MIPs. The protocol described in this paper is well-motivated and should advance our ability to diagnose radiative forcings in models but also to gain insights into physical mechanisms underpinning them. Other than the fixed land surface temperature experiments (which are lower priority), the experimental design is relatively simple and the data request is fairly modest, making RFMIP a relatively low-burden MIP that yields high-value information.

This review was jointly conducted by members of the Cloud Feedback Model Intercomparison Project (CFMIP) Scientific Steering Committee. We are highly supportive of RFMIP and recommend acceptance of this paper pending consideration of the comments below.

Signed,

Mark Zelinka, Paulo Ceppi, and Alejandro Bodas-Salcedo

**Major Comments**

Question 1 can be understood as a special case of question 2. Only after reading Section 2.2 becomes clear that Q1 is specifically aiming at a more accurate characterisation of the present-day radiative forcing. It would be good to reword Q1 to highlight the emphasis on accuracy for the present-day forcing.

Regarding the framing of research question 4 on the impact of land temperature changes for forcing and feedback estimates, the paper explains that prescribed-SST simulations only approximate ERF, because land temperatures are free and they therefore induce a radiative response. This logic makes sense, but one could also argue that land warming is rapid (like other rapid adjustments) and therefore should be counted as part of the adjustments – in which case there is no need to correct for this land response when estimating ERF. Should the wording be a bit less categorical on this issue, for example at L122–124? One could for instance say that it's not fully clear whether land warming should be treated as an adjustment, a feedback, or perhaps a mix of both. To be clear: the research to understand the impact of land temperature change on ERF is valid – it's just that the framing could be adjusted.

**Minor Comments**

- RFMIP2.0: use of decimal point seems redundant and not consistent with standard CMIP nomenclature (do you plan to run RFMIP2.x iterations?)
- L47-48: Suggest also citing Zhou et al (2023) doi:10.1029/2022GL101700 here.
- L68: Should "impacts climate sensitivity" be "impacts our observationally-based estimates of climate sensitivity"?
- L72. Here "effective climate sensitivity" is used, but "climate sensitivity" or "equilibrium climate sensitivity" is used on other parts of the manuscript. Please use consistent terminology. Also, the definition is missing, it would be worth defining it when equation (1) is introduced.
- L120–122: This way of estimating ERF is sometimes referred to as "Hansen forcing" in the literature (to distinguish it from regression-based "Gregory forcing), with a reference to Hansen et al. 1997 (10.1029/96JD03436). Suggest including that phrasing and reference here.
- L127: Given that substantial drift may be occurring in some models' piControl simulations, it may be better to enforce / recommend that the 30-year segment come from the portion of the piControl simulation near where various experiments (e.g., historical, abrupt-4xCO2) actually branch from it.
- L146: should "forcing boundaries" be "forcing boundary conditions" or just "forcings"?
- L194: The DECK acronym has already been expanded in L76.
- L212: the Fast Track now seems to be referred to as "Assessment Fast Track" or AFT (cf Dunne et al. 2025 and CMIP7 website)
- L250-254: The comparison of state-dependence of ERF to the pattern effect for feedbacks is a little off or at least incomplete, because feedbacks also exhibit state-dependence independently of the pattern effect (e.g., 10.1038/s41561-020-00649-1, 10.1002/2015GL064240, 10.1029/2020GL089074).
- L285: "an equal increase or decrease in CO2 concentration". Please clarify that you are referring to a *multiplicative* increase/decrease.
- L287: strictly speaking, rather than non-linearity aren't you referring to a departure from logarithmic behaviour?

- L317: Check reference format (B. Zhang et al); it was "Zhang et al" at L300
- L422: "has become"
- L442: It is discussed that fewer modelling groups have implemented the MODIS simulator. Given that MODIS is arguably a better instrument than the sensors that provide data to ISCCP, is less susceptible to misplacing low-level clouds at mid-levels, and has a record that is now more than 2 decades long, is it worth including more explicit encouragement to modellers to include MODIS simulator output?
- Figure 1: More details are needed in the caption. How do these individual columns sum? Is anthro equal to the sum of ghg, aer, and lu? Is 1.4xCO2 a component of ghg? It is a little awkward that only the CO2 component is labeled like this (with multiplication factor in front of it) should it just be "co2" and then the explanation is given in the caption? The parenthetical statement "scaled from a 4xCO2 simulation to approximately 1.4xCO2 in 2014" is a little hard to parse and should be rephrased for clarity. Finally, use of colors seems superfluous for this figure.
- Tables 1,3,4: The caption states "minimum of 1 ensemble member" but these are time-slice experiments; unless the spin-up is important, it would be simpler to output more years than more ensemble members. Perhaps instead state "minimum 30 years"?
- Experiment names: The current naming convention (judging from the Dunne et al paper) uses lowercase k: so e.g. piClim-p4k → piClim-p4k
- Table 3: piClim-4K-aer → piClim-p4k-aer (missing "p").